# Differing Membrane Interactions of Two Highly Similar Drug-Metabolizing Cytochrome P450 Isoforms: CYP 2C9 and CYP 2C19

**DOI:** 10.3390/ijms20184328

**Published:** 2019-09-04

**Authors:** Ghulam Mustafa, Prajwal P. Nandekar, Neil J. Bruce, Rebecca C. Wade

**Affiliations:** 1Molecular and Cellular Modeling Group, Heidelberg Institute for Theoretical Studies (HITS), 69118 Heidelberg, Germany; 2Zentrum für Molekulare Biologie der Universität Heidelberg, DKFZ-ZMBH Alliance, 69120 Heidelberg, Germany; 3Interdisciplinary Center for Scientific Computing (IWR), Heidelberg University, 69120 Heidelberg, Germany

**Keywords:** cytochrome P450, isoform, membrane protein, protein-membrane interactions, enzyme substrate specificity, mutagenesis, molecular dynamics simulation

## Abstract

The human cytochrome P450 (CYP) 2C9 and 2C19 enzymes are two highly similar isoforms with key roles in drug metabolism. They are anchored to the endoplasmic reticulum membrane by their N-terminal transmembrane helix and interactions of their cytoplasmic globular domain with the membrane. However, their crystal structures were determined after N-terminal truncation and mutating residues in the globular domain that contact the membrane. Therefore, the CYP-membrane interactions are not structurally well-characterized and their dynamics and the influence of membrane interactions on CYP function are not well understood. We describe herein the modeling and simulation of CYP 2C9 and CYP 2C19 in a phospholipid bilayer. The simulations revealed that, despite high sequence conservation, the small sequence and structural differences between the two isoforms altered the interactions and orientations of the CYPs in the membrane bilayer. We identified residues (including K72, P73, and I99 in CYP 2C9 and E72, R73, and H99 in CYP 2C19) at the protein-membrane interface that contribute not only to the differing orientations adopted by the two isoforms in the membrane, but also to their differing substrate specificities by affecting the substrate access tunnels. Our findings provide a mechanistic interpretation of experimentally observed effects of mutagenesis on substrate selectivity.

## 1. Introduction

Human cytochrome P450 (CYP) enzymes play important roles in the metabolism of drugs, steroids, fatty acids, and xenobiotics. CYPs also catalyze the conversion of some prodrugs into active drugs. Only about a dozen human CYPs metabolize 70–80% of all drugs, and these mainly belong to families CYP1, CYP2, and CYP3, and their subfamilies [1]. The human CYP2C subfamily contributes significantly to the hepatic clearance of many drugs. Although the members of the subfamily exhibit about 70% sequence similarity, they have unique substrate specificity profiles [2]. The human CYP2C subfamily consists of four isoforms: *CYP2C8*, *CYP2C18*, *CYP2C9*, and *CYP2C19* [3]. CYP 2C9 is the most highly expressed CYP protein after CYP3A4 and it is responsible for the metabolism of over 12.8% of drugs, with its substrates being mostly weak acids, such as non-steroidal anti-inflammatory drugs (NSAID) [1]. CYP 2C19 has a 10-fold lower expression level than CYP 2C9, but contributes to the metabolism of 6.8% of drugs [1], although without the specificity for acidic drugs of CYP 2C9. Nevertheless, the polymorphism of *CYP2C19* can dramatically affect drug treatments. For example, it has been observed in the treatment of *Helicobacter pylori* infections with proton-pump inhibitors that are substrates of CYP 2C19, such as omeprazole, that the therapeutic efficiency is improved in patients with a poorly metabolizing *CYP2C19* genotype due to slower drug clearance [4]. Furthermore, *CYP2C19* is important for the enzymatic activation of the antiplatelet agent, clopidogrel, to its active thiol metabolite [5,6], and loss of function in the common *CYP2C19*2* allele, which has a splicing variant leading to truncation of the protein, results in poor response to clopidogrel [7]. On the other hand, *CYP2C9* polymorphism results in reduced affinity for cytochrome P450 reductase (*CYP2C9*2*) and altered substrate specificity (*CYP2C9*3*) [8].

CYP 2C9 and CYP 2C19 have distinct substrate specificities, despite having high sequence conservation with 91.2% sequence identity (see sequence alignment in Figure 1). Crystal structures of the globular domains of the proteins have been resolved by X-ray crystallography after truncation to remove the N-terminal transmembrane (TM) domain and flexible linker sequences, as well as mutation to introduce terminal expression tags (see Figure 1). Only one crystal structure of CYP 2C19 has been resolved (Protein Data Bank (PDB) identifier 4GQS) [9], whereas a number of crystal structures of CYP 2C9 in various liganded and mutated states have been determined (currently 11 PDB entries) [10,11,12,13,14,15,16,17]. The crystal structures show that CYP 2C19 differs from CYP 2C9 at two residues in the active site: L208 and L362 in CYP 2C9 are substituted by V208 and I362 in CYP 2C19 [9,11]. Other differences are seen on the outer surface of the globular domain. The three-dimensional fold of CYP 2C19 is closer to the structure of CYP 2C8 (PDB 2NNI) [12], which shares 78% sequence identity with CYP 2C19, than to the structures of CYP 2C9 (PDB 1R9O) [11] or CYP 2C9m7 (PDB 1OG2, 1OG5) [10], despite their higher sequence identity (91.2%). The latter structures were resolved after making seven substitutions (K206E, I215V, C216Y, S220P, P221A, I222L, and I223L) in the F’–G’ loop region of CYP 2C9 for the purpose of crystallization, as this part of the protein is hydrophobic and interacts with the membrane [10]. The structure of CYP 2C9m7 differs from that of CYP 2C9 in the B–C loop, which is highly flexible in the 1R9O structure, and the conformations of the F’ and G’ helices, which are missing in the 1R9O structure. The F’–G’ region shows high structural variation amongst the crystal structures of CYP 2C9; in structures in which the protein has the wild-type sequence, the F’–G’ region is either missing (e.g., in PDB 1R9O) [11], has an extended loop conformation and a small G’ helix (PDB 5W0C) [17], or has an F’ helix followed by a loop in the G’ region interacting with a peripherally bound ligand [15]. CYP 2C9m7 also differs in the position of the sidechain of R108, which points out of the binding cavity in the CYP 2C9m7 (1OG2) structure and inside in the CYP 2C9 (1R9O) structure. The structure of CYP 2C19 shows a more than 3.0 Å C*a* atom deviation from both the CYP 2C9 (1R9O) and CYP 2C9m7 (1OG2) structures on the outer surface entrance region of the protein responsible for substrate access and selectivity. The main differences are observed in helices F, F’, G’, and G and their turns, the turn in β-hairpin 1, and the B–C loop region.

The sequence differences outside the CYP active site binding cavity may be responsible for the differential selection of drugs entering the binding pocket [18,19,20,21]. Indeed, differences in the use of the access tunnels have been suggested by mutagenesis studies on CYP 2C9/2C19 chimeras [18,19,20,21] and simulations of the globular domain of CYP 2C9 [22]. For example, CYP 2C19 selectively hydroxylates omeprazole and S-mephenytoin, whereas CYP 2C9 has little activity against these substrates. However, substitution of residues outside the binding site (I99H, S220, and P221T) at the entrance to tunnel 2b (using the nomenclature of Cojocaru et al. [23]) increased the omeprazole 5-hydroxylase activity of CYP 2C9 to a level similar to CYP 2C19 [18]. On the other hand, the E72K substitution in CYP 2C19 was shown to decrease its enzymatic metabolic activity against three tricyclic antidepressant (TCA) CYP 2C19 substrates, amitriptyline, imipramine, and dothiepin, whereas the K72E mutation in CYP 2C9 increased its metabolic activity against these compounds [21]. Most of these differences are found in the substrate recognition sites (SRS) identified by Zawaira et al. [24] (see Figure 1). Since most residues that differ between CYP 2C9 and CYP 2C19 are found in these SRS regions, we hypothesized that the sequence differences in the SRS regions and, thereby, the conformational differences observed between the two CYPs, can contribute to different protein–membrane interactions which, in turn, can lead to differences in the substrate access tunnels to the binding cavity and the product release tunnels

To investigate the effect of sequence differences between CYP 2C9 and CYP 2C19 on the protein–membrane interactions and the orientation of the protein globular domain in the membrane, we applied our optimized multiscale modeling and simulation protocol [25] to model and simulate the two proteins in a 1-palmitoyl-2-oleoyl-sn-glycero-3-phosphocholine (POPC) bilayer. We have previously applied a similar procedure to simulate CYP 2C9 in a POPC bilayer [22]. For each system, this protocol starts with building a model of the full protein in a POPC bilayer based on the crystal structure of the globular domain (see Figure 2), and then proceeds with optimizing the system to reach a converged arrangement by coarse-grained (CG) and all-atom (AA) molecular dynamics (MD) simulations (see Materials and Methods (Section 4) for details). We compared the behavior of the two proteins in the simulations and compared our results with previously reported experimental and computational data.

## 2. Results and Discussion

### 2.1. CG Simulations Show That the CYP 2C9 and CYP 2C19 Globular Domains Adopt Distinct Orientations in the Membrane Bilayer

The coarse-grained (CG) simulations carried out are listed in Appendix A. The trajectories were analyzed to assess convergence of the orientation of the globular domain and its interactions with the membrane. Different starting structures, different lengths of the flexible linker region, and different simulation parameters were tested to ensure that reliable CYP-membrane interactions and globular domain orientations were obtained. The converged positions and orientations from all sets of simulations of CG systems are given in Table 1.

Simulations of the full wild-type sequences of the two isoforms (systems S1 and S2) show different orientations from each other in the membrane. For the full-length wild-type CYP 2C9, the angles α and β (see Figure 2 and Materials and Methods for definition of these angles) ranged from 89 to 91° and 112 to 118°, respectively. For the full-length wild-type CYP 2C19, the α and β angles ranged from 97 to 100° and 134 to 137°, respectively. The axial distance of the center of mass (CoM) of the globular domain from the membrane CoM was 42–43 ± 2 Å for CYP 2C9 and 47–48 ± 2 Å for CYP 2C19. The different orientations of the two isoforms were classified into three different classes, A, A/B, and B. When the β angle was below 125°, the orientation was categorized in class A, from 125° to 130° into class A/B, and above 130° into class B. All wild-type CYP 2C9 CG systems (S1–S3) converged to class A, and all wild-type CYP 2C19 CG systems (S1–S3) converged to class B. The angles and distance values were plotted for all the CG systems in Appendix A. Thus, the two isoforms adopt distinct positions and orientations of their globular domains in the phospholipid bilayer in the CG simulations.

### 2.2. The Globular Domain Converges to the Same Orientations and Positions with Respect to the Membrane in CG Simulations of Full Length and of N-Terminally Truncated Protein

In order to ensure that the initial modelled structures did not bias the results, two separate simulations of CYP 2C9 and CYP 2C19 were performed using the N-terminally truncated form with only the globular domain (S3). In these simulations, the globular domain could explore more configurations before reaching a stable orientation. For both isoforms, the final orientations of the globular domains in the phospholipid bilayer were stabilized by insertion of the F’–G’ helices in the bilayer. Snapshots from these CG simulations are shown in Figure 3A,B. The region of the F’–G’ helices is one of the hydrophobic regions in CYPs that keep the globular domain anchored in the membrane, even after truncation of the TM-helix. In the simulations of truncated CYP 2C9, the orientation of the globular domain converged in 3.5 µs with a sharp decrease in the CoM distance of the F’–G’ helices, shown by the arrow in Figure 3D. The orientation of the CYP 2C19 globular domain converged quickly in 200 ns and remained stable throughout the simulation (Figure 3B,F). In both CYPs, after the F’–G’ helices developed contacts with the membrane, no further change was observed in the orientation of the globular domain. For both isoforms, the converged orientation of the globular domain was the same in the simulations of the globular domain only (S3) and of the full-length protein (S1, S2) (Table 1 and Figure 3C,E). The difference in the orientations of the two CYP isoforms in the membrane was maintained in the simulations of both the full length and truncated forms of the proteins. However, in the simulations of the truncated proteins (S3), the CoM distance was the same for both proteins (46 ± 2 Å). Overall, the CYP 2C9 CG simulations showed final orientations differing from CYP 2C19 despite using the same simulation parameters, water models, and protein components, whether full length or globular domain only.

### 2.3. Structural Differences in the Interfacial Residues Affect the Protein-Membrane Interactions and Globular Domain Orientations in CG Simulations

To test the effect of the initial structure of the globular domain on its positioning in the membrane, we next performed CG simulations starting with four different models of CYP 2C9 (systems: M1–M4 (see Appendix A). These full-length models of CYP 2C9 differed slightly in the side chain conformations over the whole globular domain, and more significantly in the membrane-interacting regions due to the different templates used for modeling the protein (see Appendix B). We focused on CYP 2C9 for these tests because the F’–G’ helices were missing in the crystal structure used (PDB 1R9O), and this region shows high structural variability in the crystal structures of CYP 2C9. Models M1 and M2 were built by employing two different strategies to use the template information from the structures of both CYP 2C9 and CYP 2C19, whereas M3 was built using the CYP 2C19 structure as a template, and M4 was built using a previous model [22] for the CYP 2C9 F’–G’ helices. Important differences were observed in the conformations of β-hairpin 1, the B–C loop, and the F’–G’ helices, which are critical for developing CYP-membrane interactions and thereby influence the final orientation of the globular domain in the membrane. The differences in the orientations in the CG simulations of the different models of CYP 2C9 are shown in Table 1, Appendix A, and Appendix A.

In all these CG simulations (M1–M4), the distance of the CoM of the globular domain to the CoM of the lipid bilayer increased from 43 ± 2 to 46–48 ± 2 Å. The new CoM distance value for CYP 2C9 was the same as observed for CYP 2C19. The angles α and β increased in the CG simulations using modeled structures compared to those starting with the crystal structure of CYP 2C9 (S1, S2). In the CG simulations of M3, for which the CYP 2C19 crystal structure was the template, 50% of the simulations (three out of six trajectories) showed higher angles (class B), resembling the orientation of CYP 2C19 in the membrane (Appendix A, Appendix A). The increased angle and distance values were attributed to the initial conformations of the globular domain, due to the selection of the modeling template. The CG simulations of the four CYP 2C9 models (M1–M4) indicated that it is not only the primary sequence but also the initial conformational differences in the linker, B–C loop, and F’–G’ helix regions that influence the final positioning of the CYP globular domain in the membrane. However, it should be borne in mind that the different conformations come from templates with different sequences, and that conformational preferences are dependent on sequence.

### 2.4. CG Simulations of Chimeric Mutant Models Show Sequence Differences in the Interfacial Residues Affect the Protein-Membrane Interactions and Globular Domain Orientations

To identify the residues contributing to the differing position of the globular domain in the membrane, structures of chimeric mutants of CYP 2C9 (mt2C9) and CYP 2C19 (mt2C19) were prepared by swapping residues mainly at the membrane interface in the linker (G46D), β-strand 1–2 (K72E and P73R), the B–C loop (I99H), and the F’–G’ helices (S220P and P221T) (see Appendix A and Figure 1). In CG simulations of mt2C9, two out of five trajectories converged to a CYP 2C19-like orientation (class B), one converged to an intermediate orientation (class A/B), and two converged to the same orientation as wild-type CYP 2C9 (class A). In CG simulations of mt2C19, two out of five trajectories converged to a CYP 2C9-like orientation (class A), one to an intermediate orientation (class A/B), and two retained wild-type CYP 2C19 orientations (class B), as shown in Appendix A and Appendix A. On average, the orientation changed to the intermediate class A/B for both mt2C9 and mt2C19 (see Table 1). The membrane insertion depths of the globular domain and F’–G’ region of the chimeras showed small shifts towards the depths of the other protein.

### 2.5. All-Atom MD Simulations Based on the CG Simulations Result in Stable Systems for Both Isoforms.

We next performed all-atom (AA) MD simulations to obtain refined atomic-level interactions between the membrane and the proteins. Two different initial configurations obtained from the CG simulations (S1 and S2) were used for AA simulations of each isoform. In each case, two simulations (SIM1, SIM2) were performed starting with different initial velocities assigned from the CG:S1 configuration, one for the apo from and one for a ligand-bound form, and one simulation (SIM3) was performed starting from the CG:S2 configuration.

For both proteins, the structure of the globular domain was well preserved in these simulations, as indicated by the Cα-atom root mean squared deviation (RMSD) of the globular domain with respect to the initial energy minimized structure (see Appendix A). The structures stabilized with an RMSD of about 2–2.5 Å during the simulations. Furthermore, comparison of computed and crystallographic B-factors showed that the regions of the globular domain with large fluctuations were regions with high crystallographic B-factors (see Appendix A). This was the case for both isoforms, even though the crystallographic B-factors indicated somewhat different flexible regions in the crystal structures (see Appendix A). In CYP 2C19, the HI loop, the meander region (consisting β-bulge region and the K′ helix region), and β-strand 3 (residues 460–475) had higher B-factors (see Appendix A). The presence of the membrane did not restrict the flexibility of the membrane interacting regions, such as the linker preceding the A-helix, the B–C loop, and the F’–G’ region, and strands β1-1, β1-2, β2-1, and β2-2. Indeed, partial unwinding of the G’-helix was observed in the simulations of CYP 2C19 (2C19:SIM1). The high linker flexibility contributed to higher fluctuations in the transmembrane helix angle, γ, in this particular simulation (see Table 2).

### 2.6. All-Atom MD Simulations Confirm Differences in the Positioning of the Globular Domain on the Membrane Between CYP 2C9 and CYP 2C19

The positioning of the globular domain with respect to the membrane was analyzed by calculating the heme-tilt angle in addition to the angles and distances computed for the CG simulations. The computed values are given in Table 2. The normalized angle and distance distribution plots characterizing the position of the globular domain observed in the simulations are shown in Figure 4. The simulations of CYP 2C9 showed some readjustment in the orientations from the starting configurations in which the β angle increased from about 112° to 120–127°, corresponding to remaining in class A in two cases (for the apoprotein and for the holoprotein with a substrate, the drug flurbiprofen, bound in the active site) and transitioning to class A/B in one case (SIM3), in which the globular domain structure was slightly less stable (see Appendix A). Concomitantly, the heme-tilt angle increased from 30–34° to 40–43°. The average axial distance of the globular domain CoM from the membrane CoM increased by 0.5 to 3 Å during the simulations. Compared to previous simulations conducted with the GAFF lipid force field [22], in which the globular domain CoM distance from the bilayer center decreased from 39.5 ± 2.5 Å (from CG simulations) to 34.1 ± 1.0 Å during AA simulations, the globular domain was less immersed in the membrane, facilitating the slightly lower observed values of the α and β angles. These differences can be attributed in large part to differences in both the protein and the lipid force field used. Our studies have shown that the force field used in the current work produces results in better agreement with experimental data for simulations of CYP-membrane systems, including excellent agreement with linear dichroism measurements of the heme-tilt angle of CYPs in Nanodiscs [25,26,27].

The starting structures of the two AA models of CYP 2C19 in the membrane (for the apoprotein and for the holoprotein with the inhibitor 0XV bound in the active site) varied slightly in the heme-tilt angle: 52° in the structure from CG:S1 and 46° in the structure from CG:S2. The distance of the CoM of the globular domain to the bilayer center was also different in the two starting structures: 47 Å in CG:S1 and 50 Å in CG:S2. During all three AA simulations, the globular domain fluctuated around the starting position. The angles ranged between 95 ± 5° to 106 ± 4° for α and 136 ± 7° to 149 ± 5° for β (corresponding to class B), with an increase during the simulations of the heme-tilt angle to 55 ± 7° to 61 ± 5°. The axial distance of the CoM of the globular domain from the bilayer center stayed constant at about 46 ± 2 Å in the AA simulations, independently of whether a ligand was present (SIM2) or the TM-helix was truncated (SIM3).

In summary, the differences in the orientations and interactions of the two proteins (CYP 2C9 and CYP 2C19) observed in the CG simulations were maintained and, in some cases, became more pronounced during the AA simulations. Comparison of simulations of the apo- and holo- forms of the proteins indicate that the orientation of the proteins in the membrane was not significantly affected by the presence of a compound in the active site.

### 2.7. Key Residues Contribute to Differences in the Membrane-Protein Interactions of CYP 2C9 and CYP 2C19

From the CG and AA MD simulations, we found that despite high sequence conservation (92% sequence identity), the two isoforms of the CYP2C subfamily, CYP 2C9 and CYP 2C19, maintained differences in the interactions, orientations, and degree of insertion in the membrane of the globular domain. The most important differences in residues were found in the substrate recognition sites (SRSs). For example, the SRS1′a, SRS1′b, SRS1 and SRS2,3 regions defined by Zawaira et al. [24] covered residue differences in the linker region (G46D), between strands β1-1 and β1-2 (72–73 KP-ER), in the B–C loop (I99H) and between the F’ and G’ helices (220–221 SP-PT) (see Figure 1).

We therefore analyzed the trajectories to differentiate residues interacting with the membrane head or tail regions, calculated by defining a 5 Å distance cutoff between protein and lipid head group (phosphate atoms) and hydrophobic tail, separately. The % contact time or occupancy of these residues with the membrane components in the trajectories is shown in Figure 5A. In the simulations, the interactions of the globular domains of both isoforms with the membrane were mainly developed through strands β1-1 and β1-2 (residues 64–74), and the F’–G’ region (residues 210–226) (see Figure 5A,B). However, CYP 2C9 showed further interactions with the membrane through the A-helix (residues 50–60), the B–C loop (residues 95–110) and the C-terminal β-sheet 2 (residues 370–385). The peripheral interactions developed by CYP 2C9 were similar to the hydrophobic surface identified in CYP 2C5 (residues 30–45, 60–69 after the A-helix, 376–379 in β-strand 2-1 and the F’–G’ helices) [28]. These secondary interactions, with either lipid tail or head regions, were established by the SRS regions that showed primary sequence differences in the two CYP isoforms studied here. Therefore, differences in the above-mentioned SRS regions can be crucial for CYP-membrane interactions and the orientation of the globular domain, and may lead to distinct substrate specificity.

We extended our sequence comparison to include the four main human CYP2C family members, CYP 2C8, 2C9, 2C18, and 2C19, and rabbit CYP 2C5 to examine the residues differing at the interface region (see Figure 5**C**). Sequence comparison showed that only CYP 2C19 had a positively charged residue, R73, at this position, as all other CYP2C members had P73. Furthermore, only CYP 2C9 had a positively charged residue at position 72 (K72) when compared to the other CYPs; this residue could play a role as a selectivity filter for attracting the acidic substrates preferred by CYP 2C9 and repelling basic compounds. In the F’–G’ region, the S220 and P221 residues were only found in CYP 2C9, whereas the polar residue T221 was present only in CYP 2C19. Substitution of S220P and P221A in the CYP 2C9m7 structure (PDB 1OG2) shifted the position of P221 to that of P220, as in all other CYP2C members, which resulted in a turn between the F’–G’ helices and further stabilized the G’-helix. Experimentally, it has been observed that the substitution of CYP 2C9 residues I99H, S220P, and P221T enhanced omeprazole 5′-hydroxylation activity of CYP 2C9 [18]. In another experimental study, E72K substitution in CYP 2C19 decreased the metabolic activity, whereas the K72E substitution in CYP 2C9 increased the binding affinity of tricyclic antidepressant (TCA) drugs such as imipramine [21]. Together, the analysis of primary sequence, protein-membrane orientation and interactions, and experimental findings, supports the role of different residues in SRS regions in determining the distinct orientations of the two CYPs, leading to differences in substrate access and selectivity.

### 2.8. Structural Differences Result in Different Membrane-Protein Interactions

Structural analysis of CYP 2C9 and CYP 2C19 (PDB 1R90 and 4GQS, respectively) revealed different conformations of β-sheets 1 and 2 and the B–C loop (highlighted by rings in Figure 5B and which were observed in simulations of apo and ligand-bound proteins (Appendix A). During simulations, strands β1-1 and β1-2 remained inserted in the membrane or interacted with lipid head groups in CYP 2C9, whereas these strands made far fewer contacts in CYP 2C19. The conformation and orientation of strands β1-1 and β1-2 favored the interaction of β-sheet 2 (residues 370–385) with the membrane head groups in CYP 2C9, whereas this interaction was almost completely absent in CYP2 C19 (Figure 5A).

β-sheet 1 in CYP 2C9 differed in sequence (residues K72–P73) from CYP 2C19, (residues E72–R73), as well as in conformation (see Figure 5C). The turn between strands β1-1 and β1-2 in CYP 2C9 pointed away from the globular domain towards the membrane surface (see Figure 5B). K72 in CYP 2C9 pointed towards the binding pocket and, during the simulations, its positively charged ε-amino group transiently formed a hydrogen bond with S220 in the F’–G’ helices and with the phosphate head groups of lipid molecules (see Figure 6C). The interaction with the lipid head groups resulted in the further insertion of the β1-hairpin residues into the membrane (Figure 6). K72 has been suggested to play an important role in the selection of anionic substrates in CYP 2C9, and is positioned along pathway 2b (Figure 6, a description of ligand pathways is given by Cojocaru et al. [29]) for ligand entrance into the binding pocket from the membrane [21]; it is replaced by E72 in CYP 2C19. Besides K72, the presence of P73 in CYP 2C9 favored interactions with the hydrophobic interior of the membrane (also seen in the % occupancy contact plot, Figure 5A). Thus, the K72 and P73 residues of the β-hairpin (between the β1-1 and β1-2 strands) in CYP 2C9 could be determinants of the difference in the orientation of the globular domain in the membrane compared to CYP 2C19.

In the simulations of CYP 2C19, the β1-1 and β1-2 strands showed fewer interactions with the membrane surface, which could be attributed partly to its charged residues E72 and R73 and the conformational differences observed in the crystal structure. In various studies on membrane–protein interactions, it has been found that arginine has a propensity to stay in the lipid head group region [30,31,32,33]. In CYP 2C19, the R73 sidechain pointed towards the membrane and, thereby, appeared to restrict insertion of the β-strands in the membrane. Together, the differences in the interactions with the membrane resulted in greater tilting of the distal side of the globular domain towards the membrane, resulting in higher β angles and higher heme-tilt angles, for CYP 2C19 than for CYP 2C9 (see Figure 5B).

An important difference between CYP 2C9 and CYP 2C19 was seen in the B–C loop, which differed in only one residue, residue 99 (I in CYP 2C9, H99 in CYP 2C19). The B–C loop in CYP 2C9 was highly mobile compared to CYP 2C19, in which a B’–C’ helical conformation was present. The B–C loop also differed in the side chain conformations of R105 and R108 in the two CYPs. R105 in CYP 2C19 pointed outward and showed electrostatic interactions with D224 in the G’ helix. In the CYP 2C9 crystal structure (PDB 1R9O), R105 had a different conformation and no interactions were reported with D224 due to the missing G’-helix. However, after modeling of the F’–G’ helices and simulations, similar interactions between R105 and D224 were observed in most simulations.

From the CG simulations, we identified a role for the F’–G’ helices in stabilizing the interactions and orientation of the globular domain in the membrane. The AA simulations showed differences between the two CYPs in the F’–G’ helices (S220P and P221T). P221 in CYP 2C9 was located on the outer surface of the G’-helix, which is in direct contact with the membrane and thereby favored insertion of P221 into the lipid tail region, whereas T221 at the same position in CYP 2C19 made slightly less contact with the bilayer.

Together, these results imply that not only sequence differences but also conformational differences in the regions involved in membrane-protein interactions contribute to the differences in the orientations adopted by the two isoforms in the membrane.

### 2.9. Comparison with Experiments and Previous Simulations

There are no experimental data characterizing full length CYPs and their interactions with the membrane in atomic detail. However, various experiments have been performed to study the membrane topology of CYPs and their interactions beyond the N-terminal transmembrane helix. Engineered CYP 2C9 without an N-terminal helix remained membrane-associated through the catalytic domain, as seen by atomic force microscopy (AFM) [34]. The height of the catalytic domain above the membrane was reported as 35 ± 9 Å using atomic force microscopy [35], consistent with our simulations. The binding orientation and height were also reasonably consistent with site-directed antibody-antipeptide studies [36] and the surface hydrophobicity pattern in the crystal structure of mammalian CYP 2C5 [28], the first mammalian CYP to have its 3D structure determined.

The insertion depth of the catalytic domain in the membrane was studied by tryptophan fluorescence scanning of CYP 2C2, which suggested that L36 at the start of the A’-helix, F69 at the end of the β1-1 strand, and L380 in the β2-2 strand were inserted in the lipid bilayer, whereas Y225 in the F’–G’ region is in or near the phospholipid head groups [37]. Primary sequence analysis of CYP 2C2, CYP 2C9, and CYP 2C19 showed that the CYP 2C2 residues identified by tryptophan fluorescence scanning are conserved in all three CYPs. Similar interactions were observed in our simulations of CYP 2C9 where residues L36 and L380 showed interactions with the lipid tail region (100% occupancy), while residue F69 interacted with the tail region for 40% of the simulation time (see Figure 5A). In CYP 2C19, residue L36 was buried in the membrane, while residue F69 showed interactions with the membrane tail region for 49% of the simulation time. Due to the difference in the orientation of the globular domain, strands β2-1 and β2-2 did not interact with the membrane in CYP 2C19, and therefore residue L380 remained outside the membrane. In both CYPs, the F’–G’ helices formed a strong anchoring point and residue Y225 remained buried in the membrane.

The rearrangement of the linker region in the two CYPs during the simulations to expose polar residues in the flexible region next to the TM-helix (see Figure 5A) is consistent with the observed cytoplasmic accessibility of rat CYP 2B2 in rough microsomes to an antibody raised against residues 24–38 [36]. The linker region consists of a patch of polar residues, including several positively charged residues (22–30), a hydrophobic proline-rich patch (residues 30–40), and a patch of polar residues (40–49). The linker orientation and interactions in the two CYPs, the distribution of amino acid residues in the lipid bilayer and their propensity to reside in the lipid head or tail region matched well with the hydrophobicity scale for amino acids determined by various experiments and MD simulation studies [38,39]. During simulations, the linker remained highly mobile and changed conformation to keep polar residues in the polar patches outside the membrane core, including polar residues and charged residues that were buried in the hydrophobic core of the bilayer in the initial structure.

The orientations of CYP 2C9 in the membrane in the current study using the LIPID14 forcefield matched well with previously our published work on CYP 2C9 [22] (see Table 3 for heme-tilt angles). MD simulations of CYP 2C9 in a 1,2-dioleoyl-sn-glycero-3-phosphocholine (DOPC) bilayer by Berka et al. [40,41] resulted in orientations with a higher heme-tilt angle and greater burial of the globular domain in the bilayer, with more ligand pathways leading from the heme into the bilayer. The authors used a different initial crystal structure of CYP 2C9 (PDB 1OG2), and a different procedure to model and generate starting orientations. The Berger united atom forcefield for lipids was used, which could also contribute to different membrane-protein interactions and orientations. A bilayer of DOPC is slightly thinner than a bilayer of POPC (by 0.3 Å) and has a larger area per lipid (by 4 Å^2^) [42]. These small differences might lead to slightly more tilting of the TM-helix in DOPC than POPC, but we would expect the dipping of the CYP globular domain into the bilayer to be similar for DOPC and POPC, as they have the same head group. Interestingly, however, Berka et al. observed that the globular domain of CYP 2C9 was immersed in a depression in the bilayer, surrounded by phospholipid head groups [41].

The predicted orientation of CYPs in a DOPC bilayer has been reported in the OPM (orientation of proteins in membranes) database (https://opm.phar.umich.edu/). The heme-tilt angle of the orientation of CYP 2C19 (PDB: 4GQS) reported in the OPM database is 74°, notably higher than that observed in our MD simulations (see Table 3). A similar orientation with a heme-tilt angle of 72° is given in the OPM database for the CYP 2C9 crystal structure with PDB ID 1OG5, whereas the 1R9O structure has an orientation with a heme-tilt angle of 60°. These discrepancies may have arisen because the OPM uses the crystal structure as is to predict the protein orientation in the membrane, and therefore the orientation may be affected by the missing (PDB 1R9O) or mutated (PDB 1OG5) residues in the F’–G’ loop region, or the lack of flexible linker and TM-helix residues.

## 3. Concluding Discussion

The two isoforms of the CYP2C subfamily CYP 2C9 and CYP 2C19 exhibit ~92% sequence identity, yet they show distinct substrate specificity. Since mammalian CYPs are anchored in the endoplasmic reticulum membrane by an N-terminal helix and secondary contacts from the catalytic domain, differences in the sequence and 3D structure in the membrane-interacting region in the catalytic domain can lead to different membrane-protein interactions. As it has been hypothesized that lipophilic substrates enter into the binding pockets of CYPs from the membrane core, determining the orientation of CYPs in the membrane can provide insights into differences in the opening of ligand entrance tunnels to the membrane, substrate specificity, and the mechanism of drug selectivity. Here, we have used a multiscale simulation methodology to understand the differences in primary sequence and 3D structure of two CYPs and their impact on their interactions and orientations in the membrane.

The results of multiple CG and AA simulations showed consistency and a clear tendency for the two CYPs to adopt different orientations and positions with respect to the membrane bilayer. The orientations adopted by the globular domains of the two CYPs were classified into classes A, B, or A/B (intermediate orientation). CYP 2C9 mainly adopted a class A orientation, which has lower α, β and heme-tilt angles, and CYP 2C19 adopted a class B orientation. The class B orientation is similar to the orientations observed in simulations with the same bilayer and force field for CYP 3A4 and N-terminal mutants of CYP 17A1 and CYP 19A1, with measured heme-tilt angles of about 60° [27]. CYP 2C9 thus appears to be unusual in this set of CYPs (simulated under the same conditions) in adopting an orientation with a lower heme-tilt angle. Berka et al. simulated the six major drug-metabolizing CYPs (which vary much more in sequence than the CYP 2C9/CYP 2C19 pair studied here) in a DOPC bilayer [41]. The computed heme-tilt angle varied over the six proteins between 56 ± 5° and 72 ± 6°, with CYP 2C9 having a heme-tilt angle of 61°: at the lower end of this range, although higher than found in our simulations. The tendency to a low heme-tilt angle for CYP 2C9 was also observed by Cojocaru et al. [22], with a slightly lower heme-tilt angle when an F–G loop was present instead of the F’–G’ helices. It is reasonable to expect that ligand passage may involve unwinding of the F’–G’ helices into a loop extending further into the membrane that can open up the entrance to the ligand access route from the membrane. However, we do not expect this change in conformation to result in an increase in the heme-tilt angle.

The difference in the sequence and in conformations in the SRSs near the membrane interface resulted in different orientations and insertion depths in the membrane of the globular domains of the two CYPs. A mutational swap of the key residues differing at the membrane interface in CYP 2C9 (or 2C19) in the CG simulations resulted in similar orientations (about 50% of simulation results) to the wild-type CYP 2C19 (or 2C9) orientation. Therefore, we concluded that altering a few key residues that differ in the linker, the β1-1 strand, and the F’–G’ region through mutation or a change in conformation can significantly influence the orientation and interactions of CYP-membrane systems. McDougle et al. showed that a double mutant in the F–G loop of CYP 2J2 lowered membrane insertion in MD simulations and tryptophan fluorescence studies [44]. Notably, despite the importance of the F’–G’ region, we found that mutating residues in the F–G loop alone was not sufficient to switch the orientation of CYP 2C9 to that of CYP 2C19 or vice versa. The mutation of additional residues in the β1-1 and 1-2 (K72E and P73R) and B–C loop (I99H) was necessary.

The orientation of the CYP globular domain in a membrane may be affected by simulation parameters such as the force field used and the procedure used to model and sample the conformational space of the simulated systems. We have shown previously that the force field and procedure employed here can give good agreement with experiments [25,27]. However, other factors relevant in vivo may affect the orientation of the CYP globular domain, such as homodimerization [37], heterodimerization, or the binding of redox partner proteins. Indeed, the orientational preferences of the different CYPs may affect their ability to present the proximal binding face for effective electron transfer from cytochrome P450 reductase, or to engage in CYP oligomers. The orientation of the protein also affects the access of substrates from the membrane to the active site. We have here observed the opening of tunnels to a water probe. Further exploration would require simulation of the passage of substrate molecules by standard or enhanced sampling approaches [26,29,45]. Furthermore, allosteric ligands may affect protein orientation and substrate tunnel opening, e.g., in CYP 3A4, the allosteric ligand binds at the protein-membrane interface [46].

The membrane composition may also have an influence on protein positioning in the membrane. We have here simulated the proteins in a pure POPC bilayer. Phosphatidylcholine is the main lipid component of the mammalian ER membrane [47], and POPC bilayers are often used as a simple ER mimic in in vitro studies. For example, we previously compared the heme-tilt angle computed from simulations of three different CYPs in a POPC bilayer with that measured in experiments done on these CYPs in a Nanodisc containing a POPC bilayer and found very good agreement [27]. However, the ER membrane in fact contains a variety of glycerophospholipids, as well as cholesterol and ceramide. This more heterogeneous membrane composition may affect protein positioning and dynamics as well as ligand entrance to the active site. Indeed, Navratilova et al. [48] found that the orientation of CYP 3A4 changed as the cholesterol content of a DOPC bilayer was changed, with the heme-tilt angle increasing with increasing cholesterol. The addition of cholesterol also altered the substrate access tunnel opening patterns due to interactions of the protein with the cholesterol and the ordering and thickening of the membrane due to cholesterol. Molecular simulation studies with more realistic membrane compositions, such as that employed by Park et al. in their simulation of CYP 19A1 to mimic the rat liver endoplasmic reticulum membrane [49], will be necessary to fully understand the structural and dynamics interplay between the CYP proteins, substrates, and the membrane.

In conclusion, our MD simulations demonstrate that small sequence changes at key positions can result in distinct orientations of CYP proteins in a phospholipid bilayer. These differences affect substrate access tunnels to the active site from the membrane. The differences observed for CYP 2C9 and CYP 2C19 are consistent with their differences in substrate selectivity, providing further evidence that substrate selectivity is governed by the residues lining the substrate access route as well as those in the active site.

## 4. Materials and Methods

*Preparation of structures of full-length models of CYP 2C9 and CYP 2C19*—Our simulations of CYP 2C9 (Uniprot id P11712) were based on the crystal structure of CYP 2C9 (PDB 1R9O, in the Protein Data Bank http://www.rcsb.org), resolved at 2.0 Å resolution in complex with flurbiprofen. This structure of the globular domain of CYP 2C9 was chosen because it was the highest resolution structure available that was determined with the wild-type sequence (apart from removal of the N-terminal residues 1–25 and addition of terminal expression tags). The structure had missing residues in the linker region (residues 38–42) and in the F’–G’ region (residues 214–220). The crystal structure of CYP 2C19 (Uniprot id P33261) (PDB 4GQS: chain A) in complex with the inhibitor (2-methyl-1-benzofuran-3-yl) (4-hydroxy-3,5-dimethylphenyl) methanone (Protein Data Bank chemical component 0XV) was used. It was resolved at 2.87 Å, after truncating residues 1–28 from the N-terminus and adding expression tags [9]. The missing residues in the linker and globular domain of CYP 2C9 were similar to those in CYP 2C19 as shown in Figure 1, where the sequence alignment and secondary structure were generated using the online tool ESPript3.0 (http://espript.ibcp.fr/ESPript/ESPript) [50]. Therefore, the crystal structure of CYP 2C19 was used as a template for modeling the missing linker residues and F’–G’ residues in CYP 2C9. The TM-helix (residues 3–21) and missing linker (residues 22–25) of CYP 2C9 were modeled similarly to Cojocaru et al. [22], who modeled and simulated CYP 2C9 in a POPC membrane starting from the crystal structure (PDB 1R9O). The TM-helix of CYP 2C19 was predicted to span from residues 4–20 or 3–22 by the online server Psipred (http://bioinf.cs.ucl.ac.uk/psipred/), which uses the membrane protein structure and topology (MEMSAT3) software and transmembrane protein topology prediction using support vector machines (SVM-MEMSAT) software [51]. The PredictProtein server (https://www.predictprotein.org/) [52] suggested an N-terminal alpha-helical conformation spanning residues 2–23, and this assignment was used for the simulations of CYP 2C19, along with additional simulations with assignment of the TM-helix to residues 3–21 for consistency with the simulations of CYP 2C9. The final models of each protein consisted of the crystal structure of the globular domain with modeled missing regions (we have referred to these below as systems with the letter S). For each of the proteins, 10 different starting orientations of the globular domains above the membrane were generated by changing the dihedral angles in the linker regions to generate a diverse set of initial structures with the CYP globular domain positioned to ensure that it was outside the membrane bilayer when the protein is immersed in a bilayer (see below). These structures were used for the construction of CG models.

*Preparation of additional models of CYP 2C9*—Additional CG systems of CYP 2C9 were prepared by using four modeled structures. Since the crystal structure (PDB 1R9O) of CYP 2C9 lacks the F’–G’ helices (or F–G loop), different modeling approaches with different template structures were used to assemble four structures of full-length CYP 2C9 (see Appendix B). The CG systems (M1–M4) prepared with these modeled structures of CYP 2C9 have been designated by the letter M for “models”, which differ from the CG systems indicated by S, for which crystal structures were used (with modeled missing regions only) (Appendix A).

*Modeling of chimeric mutants of CYP 2C9 and CYP 2C19*—The residues at the protein-membrane interface differing between CYP 2C9 and CYP 2C19 were substituted to create chimeric CYP 2C9/2C19 structures. The residues of CYP 2C9 substituted by CYP 2C19 residues were in the linker (G46D), β-strand 1–2 (K72E and P73R), B–C loop (I99H), and F’–G’ helices (S220P and P221T). The corresponding substitutions were also made in CYP 2C19. Five different orientations of the wild-type all-atom models (S1) were selected to make the substitution mutations, while keeping the initial orientations of the globular domain of the mutant and wild-type structures the same. These modeled mutants are referred to as mt2C9 and mt2C19.

*Preparation of coarse-grained systems*—The MARTINI CG forcefield was used for CG simulations. A similar procedure was used to generate CG models of CYP 2C9 and CYP 2C19 in a POPC bilayer in water, as described in our previous work [25]. All-atom protein models were converted to MARTINI CG models using the martinize.py script (http://cgmartini.nl) and the TM-helix was immersed in a pre-equilibrated modeled CG POPC lipid bilayer consisting of 594 POPC molecules. The MARTINI version 2.2 forcefield with the standard non-polarizable water model (NPW) was used [53]. The elastic network model was used to apply additional harmonic restraints with an elastic force constant of 500 kJ·mol^−1^·nm^−2^ and a distance cut-off of 5 to 9 Å to preserve the secondary and tertiary structure of the protein during simulation. The secondary structure information was provided in a DSSP file obtained from the DSSP server (www.cmbi.ru.nl/dssp.html).

The effects of differing linker flexibility on the final orientations of CYP 2C9 and CYP 2C19 were checked by defining two different flexible linker regions: residues 22–36 and residues 26–38. The linker was kept flexible by removing the restraints on specified residues in the elastic network. CG systems consisting of the globular domain only (S3), residues 47–490, were prepared for CYP 2C9 and CYP 2C19 to allow an unbiased conformational search of the protein orientation and to evaluate convergence of the orientations in the membrane. The CG systems were solvated using the MARTINI standard water model (NPW) (S1–S3). Tests were also performed for the MARTINI polarizable water (PW) model (S4–S5) (see SI, Table 1).

*Coarse-grained simulations*—After preparation of the CG models, several different CG simulations were performed with the Gromacs software [54]. The MARTINI standard water model (NPW) was used and the non-bonded interactions were treated with a reaction field (RF) for Coulomb interactions, and the cut-off distance for these and for van der Waals’ interactions was set to 1.1 nm. We also tested the polarizable water (PW) model with electrostatic and van der Waals interactions calculated by the Shift method (Gromacs 4.5.5), PME and a cut-off, or RF and a cut-off (using Gromacs 5.0.4), as in Mustafa et al. [25]. Similar positioning of the globular domain with respect to the membrane was obtained as that in simulations with the NPW model, but simulation times to achieve convergence were much longer with the PW model, and convergence was not always achieved.

Each simulation started with a short steepest-descent energy minimization until the maximum force on a CG particle was less than 10 kJ·mol^−1^·nm^−1^. A 40 ns equilibration simulation at a constant temperature of 310 K and pressure of 1 atm was performed in the NPT ensemble, using velocity rescale (v-rescale) and the Berendsen procedure for pressure coupling before switching to the Parrinello–Rahman barostat method for production simulations of 12–20 µs. A coupling constant of 12 ps was used to maintain semi-isotropic pressure coupling with a compressibility of 3.0 × 10^−5^. A time step of 20 fs was applied.

*Convergence of coarse-grained simulations*—The CG simulations were considered converged when no further significant changes in the orientations of the CYP globular domains above the membrane were observed. The orientation and position of the CYP globular domain was specified by the angles and distances defined previously [22,25,55] (see Figure 2). The angles were computed by defining the following vectors: v1, from the center of mass (CoM) of the backbone particles/atoms of the first four residues to the CoM of the last four residues of the I-helix; v2, from the CoM of the first four residues of the C-helix to the CoM of the last four residues of the F-helix; v3, the vector between the CoMs of the first and last four residues of the TM-helix; and the z-axis perpendicular to the membrane. The angle α was then defined as the angle between v1 and the z-axis and angle β was defined as the angle between v2 and the z-axis. Angles α and β define the orientation of the globular domain above the lipid membrane. Similarly, the TM-helix tilt angle (γ) in the lipid membrane was defined as the angle between v3 and the z-axis. The axial distances of the CoM of the globular domain (residues 50–490), the linker region (residues 22–49), and the F’–G’ helices (residues 210–220) to the CoM of the lipid bilayer were monitored during the trajectories.

*Back conversion from CG to AA models*—For each system, representative frames from the converged parts of each set of CG production runs were selected for back-conversion to an all-atom model. The representative frame was chosen to have angle and distance values within 1% of their mean value over the converged parts of the production runs [55]. The back-conversion of the POPC bilayer was performed as described in Cojocaru et al. [22], whereas the protein back-conversion was done using scripts backward.py and initram.sh, available at the MARTINI website (http://cgmartini.nl) [56]. In the absence of the heme cofactor in the CG model, conformational changes in the side chains of the heme-binding pocket residues were observed. Therefore, the globular domain (residues 50–490) from the crystal structure was superimposed on the back-mapped structure and used in subsequent AA simulations. The AA model of the globular domain contained the heme-cofactor. If there was a co-crystallized ligand in the crystal structure, it was also reincorporated in the model. The TM-helix and flexible linker region obtained from the back-conversion procedure were then connected to the globular domain, resulting in a full-length all-atom model. Finally, the all-atom model of the CYP was placed into the all-atom model of the POPC bilayer to obtain a complete CYP-membrane complex.

*All-atom molecular dynamics simulations of CYP 2C9 and CYP 2C19*—AA MD simulations were performed with two different starting orientations of the CYPs in the membrane for each CYP. Different orientations were obtained for each CYP from two different CG simulation systems, S1 and S2. AA forcefields AMBER ff14SB [57] and LIPID14 [58] were used for the protein residues and for the POPC lipids, respectively. The heme parameters were provided by D. Harris with the partial atomic charges derived from DFT calculations [59]. The ionic concentration was maintained at 150 mM using Na^+^ and Cl^−^ ions in a periodic box of TIP3P [60] water molecules. The same procedure for AA MD simulation was used as described by Cojocaru et al. [22]. The simulation protocol began with energy minimization with a decreasing harmonic force constant of 1000 to 0 kcal/mol.Å^2^ on non-hydrogen atoms of the protein and lipid residues, as described in Reference [22]. The system was then equilibrated using NAMD 2.10 [61] in a constant surface area, pressure, and temperature (NPAT) ensemble, for 1.5 ns with a gradual decrease in the harmonic restraints from 100 to 0 kcal/mol.Å^2^ on non-hydrogen atoms of the protein and lipid residues. The equilibration simulations in the NPAT ensemble were extended to 10 ns without harmonic restraints with a 1 fs integration time, keeping water molecules rigid. During subsequent production simulations, all bonds were kept rigid and the time step was increased to 2 fs. Anisotropic pressure coupling was applied, in which the cell fluctuates independently in the x, y, and z cell dimensions.

Control calculations were also performed with the GAFF lipid force field which was used in previous work [22]. The GAFF lipid forcefield requires surface tension to maintain the structural properties of the membrane bilayer, whereas the LIPID14 parameters are optimized for use without application of surface tension. We also assessed semi-isotropic pressure coupling for the simulations with the LIPID14 force field. The results show that the alternative simulation parameters gave the same class of orientation of the globular domain in the bilayer as obtained with LIPID14 and anisotropic pressure coupling. We have previously found that the combination of AMBERff14SB and LIPID14 gives better agreement with experiment than the GAFF force field and that it results in heme-tilt angles for CYPs in bilayers in excellent agreement with linear dichroism data for CYPs in Nanodiscs [27].

The orientation and position of the CYP globular domain in the AA MD simulations was characterized by computing the same angles and distances as for the CG MD simulations. In addition, the heme-tilt angle, the angle between the heme plane defined by the four nitrogen atoms coordinating the iron and the z-axis, was computed (see Figure 2). VMD (www.ks.uiuc.edu/Research/vmd/) [62] was used for the analysis and to generate the molecular graphics figures. Tunnels accessible to a water molecule probe were computed using the MOLEonline webserver (mole.upol.cz/) [63] with default parameters.

## Figures and Tables

**Figure 1 ijms-20-04328-f001:**
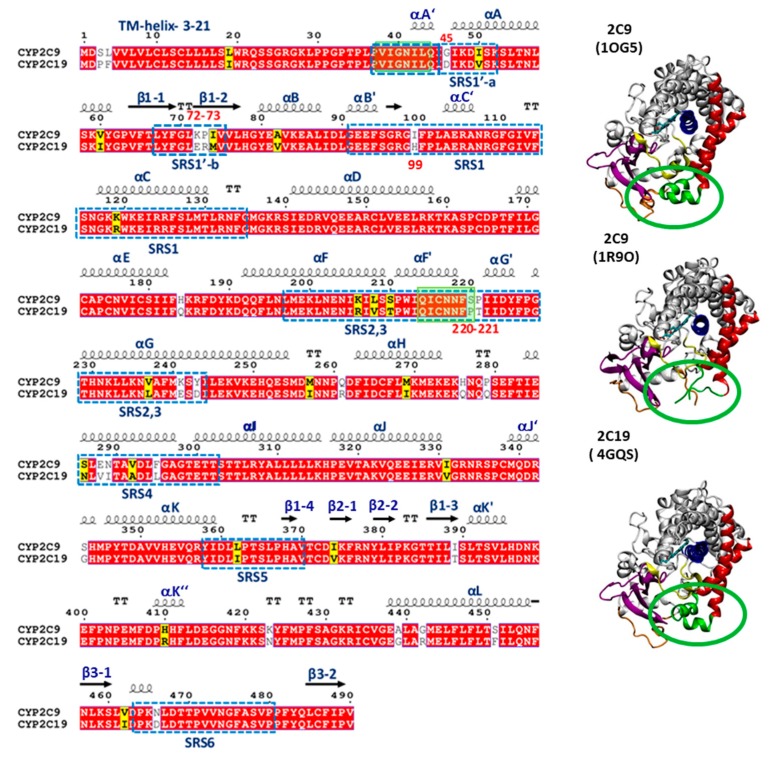
(Left) Sequence alignment of CYP 2C9 and CYP 2C19. Identical residues are shown with a red background, similar residues with a yellow background, and differing residues with a white background. The secondary structure in the crystal structure of CYP 2C19 (PDB 4GQS) is indicated by arrows for β-strands, springs for α-helices, and ‘TT’ for turns; long loops are unmarked. The substrate recognition sites (SRS) are shown by blue dashed line boxes. The residues in the globular domain differing at the membrane interface are highlighted by red numbers. The missing regions in the crystal structure (PDB 1R9O) of the globular domain of CYP 2C9 are shown by transparent green boxes. (Right) Cartoon representations of the crystal structures of CYP 2C9m7 (PDB 1OG5), CYP 2C9 (PDB 1R9O), and CYP 2C19 (PDB 4GQS), showing the structural differences in the F’–G’ region highlighted by the green rings, the heme in stick representation, and key secondary structure elements colored as follows: β-strand regions in magenta, the B–C loop in yellow, the F and G helices in red, the F’–G’ helices/loop in green, the I-helix in blue, and the linker in orange. The active site is lined by the heme and the I-helix.

**Figure 2 ijms-20-04328-f002:**
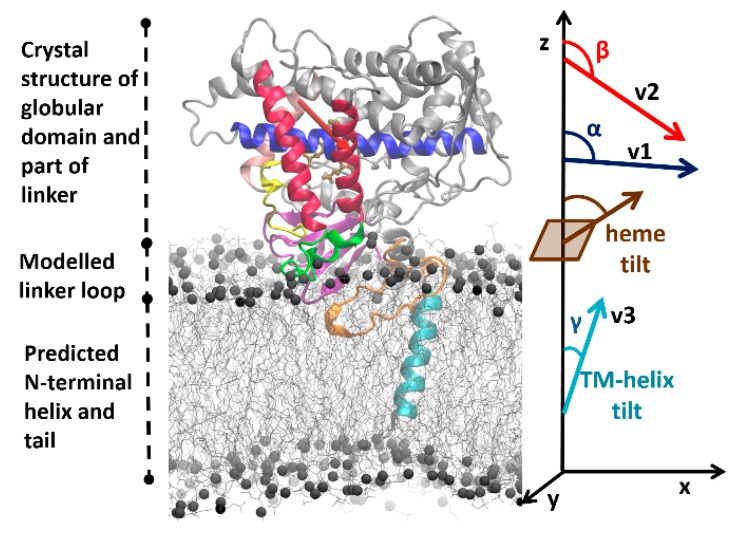
Initial model of human cytochrome P450 (CYP) 2C9 showing its three domains and the initial information on which modeling and simulation of its arrangement in the phospholipid bilayer was based. The crystal structure (PDB 1R9O) of the globular domain (residues 50–490) and part of the linker region (residues 37–49) are shown in cartoon representation. Secondary structure predictions indicate the length of the N-terminal transmembrane (TM)-helix (cyan, residues 3–21). These two components are connected by a modeled linker loop of unknown conformation (orange, residues 22–36). The flexible C-terminal tail (residues 491–492) was not included in the model. The F’–G’ helices (residues 210–220) were not observed in the structure and were modeled from the crystal structure of CYP 2C19 (PDB 4GQS). Important secondary structure elements in the globular domain are colored as follows: β-strand region in magenta, F and G helices in red, I-helix in blue, B–C loop in yellow, and F’–G’ helices in green. The heme is shown in brown stick representation. Experimentally, it is known that the globular domain interacts with the bilayer (shown in grey line representation with grey spheres representing the phosphorous atoms) and, during the coarse-grained (CG) simulations, it approached and dipped into the bilayer. The heme tilt angle and the angles α and β defining its orientation in the bilayer are shown on the right, along with the definition of the TM-helix tilt angle (γ), and the vectors (v1 along the I-helix , v2 shown by the red arrow from the C to the F helix, and v3 along the TM-helix) computed to define these angles; the definitions of these angles are given in the Materials and Methods section.

**Figure 3 ijms-20-04328-f003:**
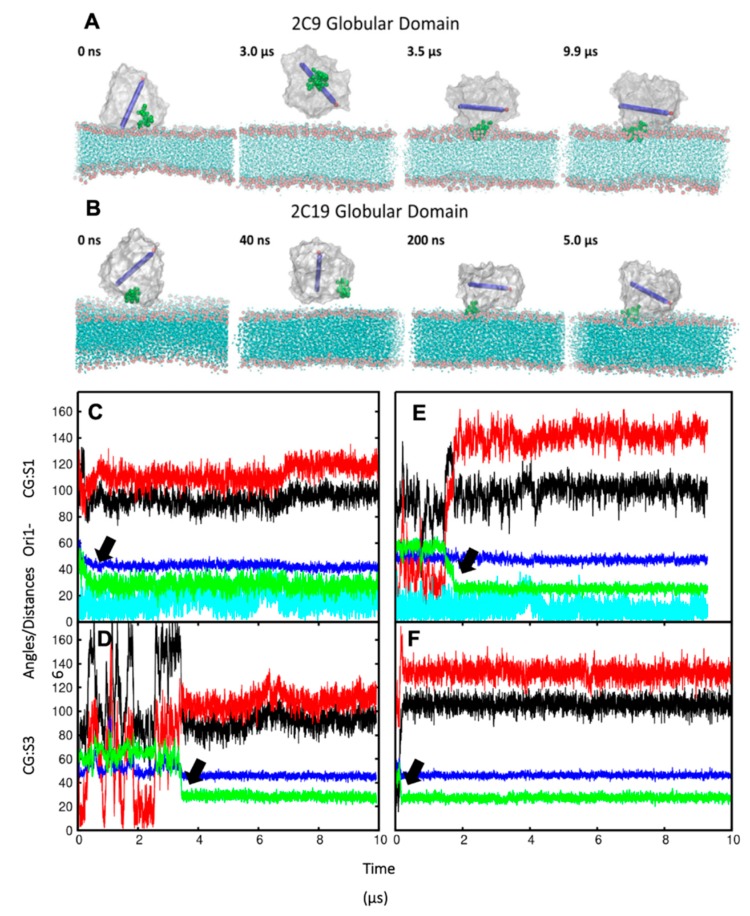
(**A**,**B**) Snapshots from CG simulations of globular domains (S3) of (**A**) CYP 2C9 and (**B**) CYP2 C19 showing exploration of different orientations followed by convergence to the same orientation as observed for CG simulations of the full-length wild-type proteins. The globular domain is shown with a silver surface representation, with the F’–G’ helices shown as green VDW spheres and the I-helix (residues 286–316) shown as a blue cylinder with an arrow and a red sphere at the C-terminal end. The 1-palmitoyl-2-oleoyl-sn-glycero-3-phosphocholine (POPC) bilayer is shown in cyan, with the phosphate atoms in the head groups shown as red spheres. (**C**–**F**) Convergence of the orientation and position of the globular domain during CG simulations of CYP 2C9 (**C**,**D**) and CYP 2C19 (**E**,**F**). The angles (°) and distance (Å) values vs time (μs) are shown for selected trajectories from CG systems: (**C**,**E**) S1 (full-length proteins); (**D**,**F**) S3 (globular domain only). The angles α (black), β (red), and the TM-helix tilt angle (cyan) (defined in Figure 2 and Materials and Methods) are shown along with the axial distances of the bilayer CoM to the CoM of the globular domain (blue) and the F’–G’ helices (green). The thick black arrows point to the decrease in the distance of the F’–G’ helices from the membrane center, which is coincident with convergence to stable orientations.

**Figure 4 ijms-20-04328-f004:**
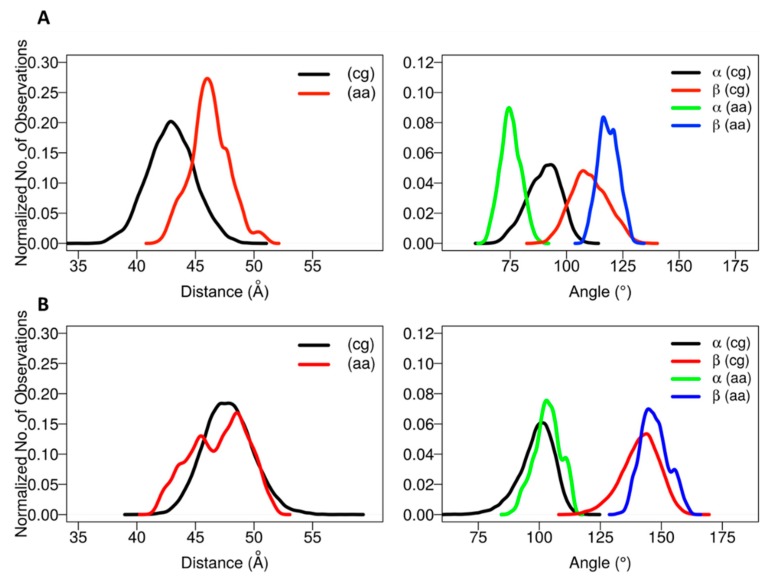
Plots of the distance and angle distributions defining the positioning of the globular domain with respect to the membrane in CG and AA MD simulations of CYP 2C9 (**A**) and CYP 2C19 (**B**). The globular domain of CYP 2C9 tended to be more immersed in the membrane than CYP 2C19 and the two adopted distinct orientations: class A (angle β < 125°) for CYP 2C9, and class B (angle β > 130°) for CYP 2C19.

**Figure 5 ijms-20-04328-f005:**
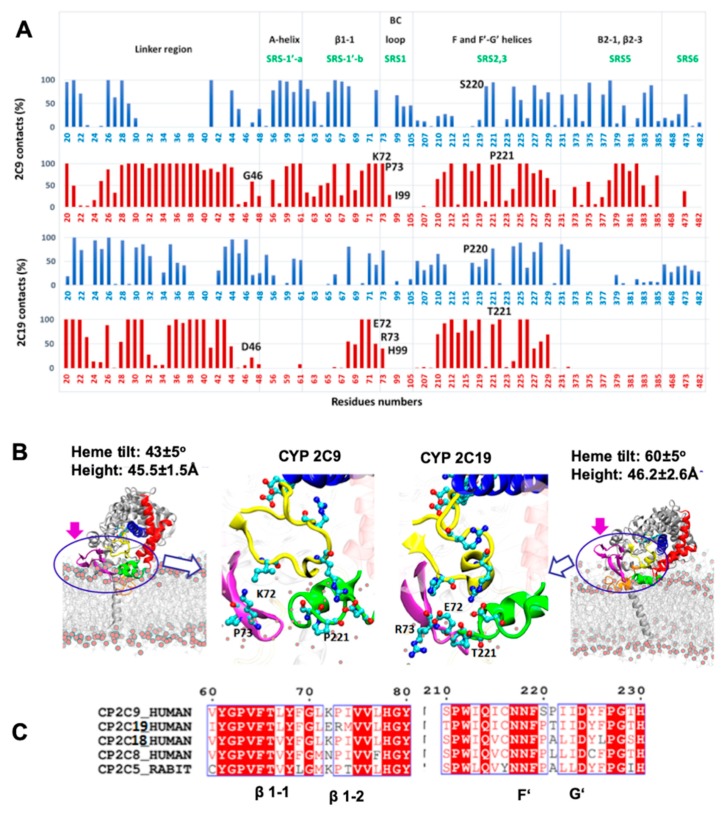
Differences in the arrangements of the CYP 2C9 and CYP 2C19 residues at the membrane interface. (**A**) Residues in contact with the lipid head group (blue) and tail region (red) during the AA MD simulations of CYP 2C9 (above) and CYP 2C19 (below). The percentage of snapshots in which a contact was present is shown on the y-axis, and the residues interacting with the membrane are given on the x-axis. The secondary structures and substrate recognition sites are shown on the top. The residues differing in the interacting regions between CYP 2C9 and CYP 2C19 are labeled. (**B**) For CYP 2C9 (left) and CYP 2C19 (right), the last frames from AA MD simulations of the apo form (SIM1) are shown for the full system and for the membrane interface region. The protein is shown in cartoon representation with selected side chains in ball-and-stick representation colored by atom type with cyan carbons. The linker is shown in orange, β-sheets 1 and 2 in magenta, the B–C loop in yellow, the F and G helices in red, the F’ and G’ helices in green, the central I-helix in blue, and the heme and key residues in cyan licorice representation. The POPC bilayer is shown in grey line representation with phosphate atoms as red spheres. The magenta arrows indicate differences in β-sheet 2 (residues 370–380). (**C**) Part of a sequence alignment of human CYP2C subfamily members (2C9, 2C19, 2C18, 2C8) and rabbit CYP 2C5. The conserved residues are shown in red for similar residues or with a red background and in white for identical residues. The residues outside the blue boxes differed amongst the aligned CYPs.

**Figure 6 ijms-20-04328-f006:**
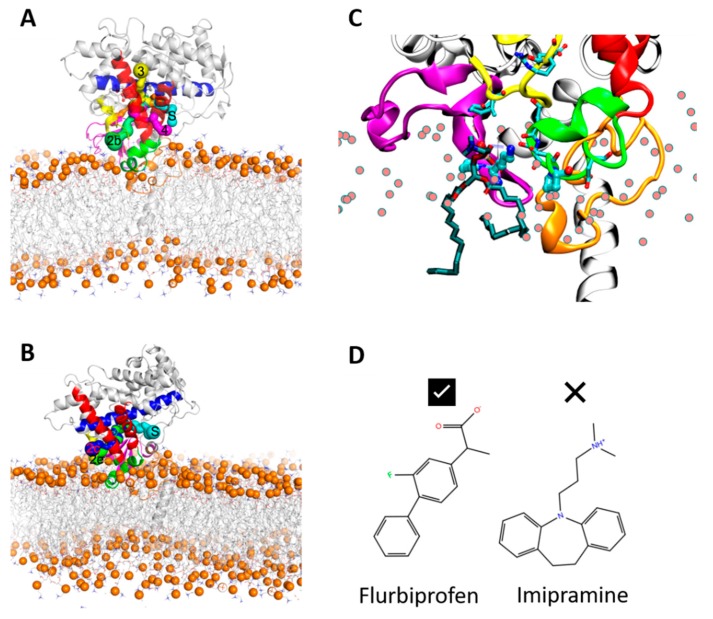
Initial (**A**) and final (**B**) snapshots of the AA MD simulation (SIM1) of the apo form of CYP 2C9, showing tunnels accessible to a water molecule probe between the active site and the protein surface. Tunnel 2b (green) connects the active site and the membrane and is present in both snapshots, as is tunnel S (cyan). Tunnel 3 (yellow) is present in the initial snapshot and tunnel 2c (blue) is present in the final snapshot. The protein and bilayer are shown with the same color scheme as in Figure 5. (**C**) Close-up view of the entrance to the 2b tunnel showing how the phosphate group of a phospholipid molecule (shown in stick representation colored by atom type with cyan carbons) makes a hydrogen bond with the amino group of K72 (all other lipid molecules are represented by spheres for the phorphorous atoms only; the protein is represented and colored as in Figure 5). This interaction is important for pulling the phospholipid molecule somewhat out of the membrane towards the tunnel to the active site. This motion leads to partial opening of the β-sheet and the F’–G’ regions; further opening would be required for a substrate molecule to access the active site. (**D**) K72 may interact analogously with acidic substrates, such as flurbiprofen, a drug that is a substrate of CYP 2C9 (left), and may repel basic substrates such as the tricyclic antidepressant (TCA) drug, imipramine, which is a substrate of CYP 2C19 (right).

**Table 1 ijms-20-04328-t001:** Results of CG simulations of CYP 2C9 and CYP 2C19 for the full length (S1, S2) and truncated (S3) wild-type proteins (based on crystal structures 1R9O and 4GQS, respectively), various models of full length CYP 2C9 (M1–M4, based on structures Model 1–4), and the mutant chimeras, mt2C9 and mt2C19. The systems simulated are listed in Appendix A. The mean and standard deviation values are given for parameters defining the position of the globular domain with respect to the membrane: angles α and β (defining the orientation, as shown in Figure 2 and class (A, A/B, or B)) and the axial distances of the center of mass (CoM) of the bilayer from the CoM of the linker, the F’–G’ region, and the globular domain, respectively. The simulations each had an average duration of 10 µs, and the parameters were computed for snapshots from the last 9 µs collected at intervals of 1 ns.

CYP Systems	Angles (°)		Distances (å)			No. of Simulations	Class
	α	β	Linker	F’–G’	Globular		
2C9:S1	89 ± 7	112 ± 7	25 ± 4	28 ± 4	43 ± 2	10	A
2C9:S2	91 ± 8	118 ± 14	26 ± 4	27 ± 4	42 ± 2	5	A
2C9:S3 ^1^	92 ± 7	109 ± 7	-	28 ± 2	46 ± 2	1	A
2C9:M1	94 ± 6	119 ± 8	20 ± 2	25 ± 2	46 ± 2	6	A
2C9:M2	92 ± 8	120 ± 12	19 ± 2	25 ± 2	46 ± 2	6	A
2C9:M3	95 ± 6	138 ± 6	22 ± 2	24 ± 2	48 ± 2	6	B
2C9:M4	85 ± 9	106 ± 9	25 ± 2	26 ± 3	47 ± 2	5	A
mt2C9	98 ± 7	129 ± 10	27 ± 4	29 ± 4	44 ± 2	5	A/B
C19:S1	100 ± 7	137 ± 10	21 ± 2	26 ± 2	47 ± 2	10	B
2C19:S2	97 ± 8	137 ± 12	20 ± 2	25 ± 2	48 ± 2	5	B
2C19:S3 ^1^	106 ± 5	133 ± 6	-	27 ± 2	46 ± 2	1	B
mt2C19	95 ± 8	127 ± 13	19 ± 2	25 ± 3	46 ± 2	5	A/B

^1^ Systems containing only the CYP globular domain, residues 47–490 for both proteins, without the TM and linker regions.

**Table 2 ijms-20-04328-t002:** Positioning of the CYP globular domain with respect to the membrane in the AA MD simulations of CYP 2C9 and CYP 2C19. Values of the mean and standard deviations of angles (as defined in Figure 2) and distances characterizing the positioning of the globular and TM domains were calculated for the last 50 ns of all-atom simulations, with different starting configurations from CG simulations and with different initial velocities assigned. The orientations of the globular domain were assigned to class A, A/B, or B.

AA MD Simulation	Residues		Angles (°)				Globular Domain-Bilayer Distance (Å)	Class	Time (ns)
	TM-Helix	Flexible Linker	α	β	γ	Heme-tilt			
2C9:CG:S1	3–21	22–36	91.9	111.9	17.6	30.2	45.0	A	10,000 ^2^
2C9:SIM1	3–21		74.8 ± 4.3	119.9 ± 4.5	11.9 ± 5.3	43.2 ± 4.8	45.5 ± 1.5	A	216.88
2C9:SIM2 ^1^	3–21		95.9 ± 4.4	123.3 ± 4.9	13.5 ± 4.1	39.8 ± 4.9	48.3 ± 2.3	A	156.1
2C9:CG:S2	3–21	26–38	90.5	111.8	13.9	33.8	42.3	A	10,000 ^2^
2C9:SIM3	3–21		86.6 ± 4.1	126.8 ± 3.0	5.9 ± 2.7	40.1 ± 5.5	44.2 ± 0.9	A/B	50.6
2C19:CG:S1	2–23	26–38	99.6	135.3	13.0	52.4	46.7	B	10,000 ^2^
2C19:SIM1	2–23	--	106.3 ± 4.2	148.6 ± 5.1	25.4 ± 7.8	60.5 ± 4.5	46.2 ± 2.6	B	108.4
2C19:SIM2 ^1^	2–23	--	97.0 ± 5.0	140.1 ± 4.2	25.3 ± 4.6	58.1 ± 5.3	45.8 ± 1.6	B	113.4
2C19:CG:S2	3–21	22–36	99.5	133.3	10.2	45.9	50.3	B	10,000 ^2^
2C19:SIM3	3–21	--	94.9 ± 4.8	135.8 ± 6.6	17.2 ± 4.1	55.4 ± 6.5	46.0 ± 1.6	B	95.2

^1^ Simulated with a ligand in the active site (all other simulations were for the apoproteins). For CYP 2C9, the ligand was flurbiprofen from the crystal structure, and for CYP 2C19, it was the inhibitor from the crystal structure (Protein Data Bank chemical component 0XV). ^2^ CG simulations were run for an average of 10 µs. The angles were computed for the representative structure that was selected for starting AA MD simulations.

**Table 3 ijms-20-04328-t003:** Comparison of the computed heme-tilt angle from AA MD simulations of CYP 2C9 and CYP 2C19 with previously reported values. The heme-tilt angle is the angle between the heme plane and the membrane normal (see Materials and Methods and Figure 2).

Source	Reference	Lipid/Force Field	Protein PDB ID	Heme-Tilt Angle (°)
	**CYP 2C9**			
**MD Simulation**	Current Study	POPC/LIPID14	1R9O	40–43 ± 5
	[22]	POPC/GAFF lipid	1R9O (1)	44 ± 4
			1R9O (2)	41 ± 4
	[40]	DOPC/Berger	1OG2	55 ± 5
	[41]	DOPC/Berger	1OG2	61 ± 4
**OPM Database**	[43]	DOPC/OPM	1R9O	59.8
			1OG5	71.9
	**CYP 2C19**			
**MD Simulation**	Current Study	POPC/LIPID14	4GQS	55–61 ± 5
**OPM Database**	[43]	DOPC/OPM	4GQS	74.0

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
