# Peer review of "Differing Membrane Interactions of Two Highly Similar Drug-Metabolizing Cytochrome P450 Isoforms: CYP 2C9 and CYP 2C19"

_ijms, 2019, doi:10.3390/ijms20184328_

Round 1

Reviewer 1 Report

In this paper, the authors present their findings on the use of modeling and simulation to evaluate the infuennce of sequence differences in 2 CYP isoforms on the 'protein-membrane interactions and the orientation of the protein globular domain in the membrane'. The work also builds on previously publicised wok of the authors. Through the modeling process, the authors are able to largely, offer interpretations of experimentally observed differences in the substrate selectivity of CYP2C19 and CYP2C9.

Minor suggestion:

Ensure consistency in using abbreviated names for cytochrome P450, eg CYP 2C19 versus CYP2C19 On line 44: The expression of CYP 2C19 is increased in response to treatment of Helicobacter pylori (H. pylori) infections with proton-pump inhibitors, such as omeprazole, lansoprazole, and pantoprazole,
thereby increasing drug clearance". First, this may suggest that H pylori has anything to do with the PPI effect on CYP. Regardless of the indication of PPI use, I think the point is between the drugs and the enzyme. Secondly, how accurate is this statement? Kindly provide reference to back it. It is well known that PPIs like omeprazome are inhibitors of CYP2C19. Not sure if they are inducers too, as implied by the statement. Line 67: The statement appears ambiguos. What does 'it' in 'in structures in which it has the' refers to? Consider revising the statement. Line 83: "On the other hand, the E72K substitution in CYP 2C19 decreased tricyclic antidepressant (TCA) activity". There are differences in the level of involvement of CYP2C19 and CYP2D6 in the metabolism of the individual tricyclic antidepressants. In the reference cited, the findings applied to  amitriptyline, imipramine, and dothiepin, which are CYP2C19 substrates. Line 106, and other places: It is generally recommended to use passive reporting, and avoid 'first person' attributions.  

Author Response

Response to Reviewer 1 Comments

Point:

In this paper, the authors present their findings on the use of modeling and simulation to evaluate the infuennce of sequence differences in 2 CYP isoforms on the 'protein-membrane interactions and the orientation of the protein globular domain in the membrane'. The work also builds on previously publicised wok of the authors. Through the modeling process, the authors are able to largely, offer interpretations of experimentally observed differences in the substrate selectivity of CYP2C19 and CYP2C9.

Minor suggestion:

Ensure consistency in using abbreviated names for cytochrome P450, eg CYP 2C19 versus CYP2C19.

Response:

We have revised the manuscript to ensure that the abbreviation of CYP 2C19 or CYP 2C9 is always used when referring to the respective cytochrome P450 proteins.  The gene names of CYP2C9 and CYP2C19 are only used when the gene is referred to, e.g. when referring to the isoforms of the human CYP2C family.

Point:

On line 44: The expression of CYP 2C19 is increased in response to treatment of Helicobacter pylori (H. pylori) infections with proton-pump inhibitors, such as omeprazole, lansoprazole, and pantoprazole,

thereby increasing drug clearance". First, this may suggest that H pylori has anything to do with the PPI effect on CYP. Regardless of the indication of PPI use, I think the point is between the drugs and the enzyme. Secondly, how accurate is this statement? Kindly provide reference to back it. It is well known that PPIs like omeprazome are inhibitors of CYP2C19. Not sure if they are inducers too, as implied by the statement.

Response:

We have modified this sentence to avoid misunderstanding and improve clarity.  The revised text is as follows:

“Nevertheless, the polymorphism of CYP2C19 can dramatically affect drug treatments, as has been observed for the treatment of Helicobacter pylori infections with proton-pump inhibitors that are substrates of CYP 2C19, such as omeprazole, in which the therapeutic efficiency is improved in patients with a poorly metabolizing CYP2C19 genotype due to slower drug clearance [4]. On the other hand, CYP2C9 polymorphism results in reduced affinity for cytochrome P450 reductase (CYP2C9*2) and altered substrate specificity (CYP2C9*3) [5]. 

 Point:

Line 67: The statement appears ambiguos. What does 'it' in 'in structures in which it has the' refers to? Consider revising the statement.

Response:

We have revised this sentence to:

“…in structures in which the protein has the wild-type sequence, the F'-G' region is either missing (e.g. in PDB 1R9O) [8], has an extended loop conformation and a small G’ helix (PDB 5W0C) [14], or has an F’ helix followed by a loop in the G’ region interacting with a peripherally bound ligand (PDB 5XXI).”

Point:

Line 83: "On the other hand, the E72K substitution in CYP 2C19 decreased tricyclic antidepressant (TCA) activity". There are differences in the level of involvement of CYP2C19 and CYP2D6 in the metabolism of the individual tricyclic antidepressants. In the reference cited, the findings applied to  amitriptyline, imipramine, and dothiepin, which are CYP2C19 substrates.

Response:

We have revised this sentence to:

“On the other hand, the E72K substitution in CYP 2C19 was shown to decrease its enzymatic metabolic activity against three tricyclic antidepressant (TCA) CYP 2C19 substrates, amitriptyline, imipramine, and dothiepin, whereas the K72E mutation in CYP 2C9 increased its metabolic activity against these compounds.”

Point:

Line 106, and other places: It is generally recommended to use passive reporting,  and avoid 'first person' attributions. 

Response:

There are differing opinions on whether scientific papers should be written using the passive voice or the active first person voice. We have used the first person in this paragraph to clearly distinguish the original work described in this manuscript from prior work discussed in the preceding paragraphs.

Reviewer 2 Report

Mustafa et al. have described multiscale modeling and simulation studies to suggest differing membrane interactions of CYP2C9 and CYP2C19. The present work is very interesting because while the functional activity of these isoforms in well understood, the mechanistic characterization of structural orientation is not well studied. Authors identified amino acid residues in 2C9 (K72, P73, and I99) and 2C19(E72, R73 and H99) 25 2C19) at the protein-membrane interface that contributes to the different orientations of the two isoforms in the membrane and their differing substrate specificity by affecting the substrate access tunnels. 

Minor comments: 

Fig 3: What I don't comprehend is why the enzyme structure is disjointed at 3.0 microsec for 2C9 and 40 nanosec for 2C19.

Fig 6. The labeling of the figure should be addressed. It's very difficult to see the numbers used to define tunnel entry points. I would suggest changing to an alternate color instead of black for labeling. Else authors can consider using arrows to point the location on the enzyme structure. 

Author Response

Response to Reviewer 2 Comments

Point:

Mustafa et al. have described multiscale modeling and simulation studies to suggest differing membrane interactions of CYP2C9 and CYP2C19. The present work is very interesting because while the functional activity of these isoforms in well understood, the mechanistic characterization of structural orientation is not well studied. Authors identified amino acid residues in 2C9 (K72, P73, and I99) and 2C19(E72, R73 and H99) 25 2C19) at the protein-membrane interface that contributes to the different orientations of the two isoforms in the membrane and their differing substrate specificity by affecting the substrate access tunnels.

Minor comments:

Fig 3: What I don't comprehend is why the enzyme structure is disjointed at 3.0 microsec for 2C9 and 40 nanosec for 2C19.

Response:

In Fig 3a and 3b, we show snapshots at these time points (3 microsec and 40 ns) to represent the structures before the globular domain moved closer to the membrane and the F’-G’ helices immersed in the bilayer, after which point there was little change in the positioning of the globular domain with respect to the bilayer, as can be seen from the plots in 3e and 3f. The time taken to reach a converged position in the bilayer varied in the different coarse grained simulations as they were started with different initial positions and orientations of the globular domain.

Point:

Fig 6. The labeling of the figure should be addressed. It's very difficult to see the numbers used to define tunnel entry points. I would suggest changing to an alternate color instead of black for labeling. Else authors can consider using arrows to point the location on the enzyme structure.

Response:

We have edited the legend to make the identity of the tunnels clearer by inserting the following text:

“Initial (a) and final (b) snapshots …. Tunnel 2b (green) connects the active site and the membrane and is present in both snapshots, as is tunnel S (cyan). Tunnel 3 (yellow) is present in the initial snapshot and tunnel 2c (blue) is present in the final snapshot.”

Reviewer 3 Report

The author are providing a very careful MD simulation study on two human cytochrome P450 enzymes, which share ~90 % sequence identity, but differ strongly in their structure. They provide detailed mechanistic interpretation of experimentally observed effects of mutagenesis on substrate selectivity. As the study sounds very convincing I have just a few minor comments.

1)     In Figure a structure 2C9 (1R9O) is shown, but it does not get clear what it is relevant for. It’s missing the helical green part, but it’s not mentioned detailed in the text.

2)     Flurbiprofen and imipramine were somehow used for simulation, but also for some affinity studies prior to an after mutation. But it the results part a detailed explanation on the two drugs and their roles is missing.

3)     Table 1 shows two structures only with 1 number of simulations. How can you get standard deviations in these cases?

4)     Labelling in Figure 5 is shifted.

Author Response

Response to Reviewer 3 Comments

Point:

The author are providing a very careful MD simulation study on two human cytochrome P450 enzymes, which share ~90 % sequence identity, but differ strongly in their structure. They provide detailed mechanistic interpretation of experimentally observed effects of mutagenesis on substrate selectivity. As the study sounds very convincing I have just a few minor comments.

1)     In Figure a structure 2C9 (1R9O) is shown, but it does not get clear what it is relevant for. It’s missing the helical green part, but it’s not mentioned detailed in the text.

Response:

In Figure 1, we show two crystal structures of CYP 2C9 and one of CYP 2C19. We highlight the differences between the structures of CYP 2C19 by the green ellipses. The structure 1R9O is missing the F’ and G’ helices shown in green in the 1OG5 structure.  We mention this in the legend to Figure 1: “….showing the structural differences in the F’-G’ region highlighted by the green rings….. the F’-G’ helices/loop in green…”.  We show the crystal structure of 1R9O here as it was used to provide the initial coordinates for the simulations described in this work.   The crystal structures shown in Figure 1 are referred to in the main text on line 53 and crystal structures are discussed in the remainder of this paragraph.

Point:

2)     Flurbiprofen and imipramine were somehow used for simulation, but also for some affinity studies prior to an after mutation. But it the results part a detailed explanation on the two drugs and their roles is missing.

Response:

We carried out all-atom MD simulations of the apoproteins and of CYP 2C9 with the substrate  flurbiprofen, and of CYP 2C19 with the inhibitor (2-methyl-1-benzofuran-3-yl)-(4-hydroxy-3,5-dimethylphenyl)methanone,  abbreviated as 0XV in the crystal structure of its complex (PDB 4GQS). These compounds were used as they were present in the crystal structures of the respective CYPs.  We did not simulate any complexes with imipramine. We show flurbiprofen and imipramine in Figure 6d as example substrates for the two CYPs that are experimentally well characterized.

We have edited and added the following sentences to the main text and the Figure 6 legend regarding the ligands simulated:

Figure 6 legend:

(d) K72 may interact analogously with acidic substrates, such as flurbiprofen, a drug that is a substrate of CYP 2C9 (left), and may repel basic substrates such as the TCA drug, imipramine, which is a substrate of CYP 2C19 (right).

Results:

l281: “….corresponding to remaining in class A in two cases (for the apoprotein and for the holoprotein with a substrate, the drug flurbiprofen, bound in the active site) and….”

l299: “…The starting structures of the two AA models of CYP 2C19 in the membrane (for the apoprotein and for the holoprotein with the inhibitor 0XV bound in the active site) vary slightly in the heme-tilt angle…”

l308: “In summary, the differences in the orientations and interactions of the two proteins (CYP 2C9 and CYP 2C19) observed in the CG simulations were maintained and, in some cases, became more pronounced during the AA MD simulations. Comparison of simulations of the apo- and holo- forms of the proteins indicate that the orientation of the proteins in the membrane is not significantly affected by the presence of a compound in the active site.”

Point:

3)     Table 1 shows two structures only with 1 number of simulations. How can you get standard deviations in these cases?

Response:

Table 1 shows the results of CG simulations for two proteins. The standard deviations are computed for a set of snapshots collected from the last 9 microsec of each simulation.  These snapshots were collected at intervals of 1ns.

For clarity, we have edited the last sentence describing Table 1 at l159:

“The simulations each had an average duration of 10 µs and the parameters were computed for snapshots from the last 9 µs collected at intervals of 1 ns.”

Point:

4)     Labelling in Figure 5 is shifted.

Response:

Labels A, B, and C of the figure parts were misplaced and have been corrected in the revision.

Reviewer 4 Report

This is a well written manuscript with clearly described results and adequate discussion. The authors performed different sorts of calculations on two different  cytochrome P450 enzymes, namely CYP 2C9 and CYP 2C19 and showed the different orientations of their domains cause changes in the interactions with the membrane and dictate the substrate specificity. 

My main concern is how relevant this study with the choice of POPC bilayer, when it is known that lipid composition of ER (to which CYPs are linked) is rather complex. It contains various glycerophospholipids: phosphatidylcholine phosphatidylethanolamine, phosphatidylinositol, phosphatidylserine but also , cholesterol and ceramide. The authors already mentioned the impact of cholesterol presence in their discussion. The discussion should be extended to describe the plausibility of the chosen setup. Also, the 2.9 chapter can be combined with the discussion.

There are a few minor issues: in lines 477-480 a degree sign is not in superscript. 

Also there is different spelling through the manuscript of enzymes: sometimes it is CYP 2C9 and sometimes CYP2C9. 

It would be good also in the main text to mention the quality of pdb models which were used to generate the starting models.

Author Response

Response to Reviewer 4 Comments

Point:

This is a well written manuscript with clearly described results and adequate discussion. The authors performed different sorts of calculations on two different  cytochrome P450 enzymes, namely CYP 2C9 and CYP 2C19 and showed the different orientations of their domains cause changes in the interactions with the membrane and dictate the substrate specificity.

My main concern is how relevant this study with the choice of POPC bilayer, when it is known that lipid composition of ER (to which CYPs are linked) is rather complex. It contains various glycerophospholipids: phosphatidylcholine phosphatidylethanolamine, phosphatidylinositol, phosphatidylserine but also , cholesterol and ceramide. The authors already mentioned the impact of cholesterol presence in their discussion. The discussion should be extended to describe the plausibility of the chosen setup.

Response:

We appreciate the reviewer’s concern and note that advances in simulation methods and force fields will enable the detailed investigation CYPs in more realistic membranes in future work. POPC is often used to represent the ER membrane in experiments and simulations because phosphatidylcholine accounts for over half the lipid component of the mammalian ER membrane (see e..g. van Meer, G., Voekler, DR., Feigenson, GW. Membrane lipids: Where they are and how they behave Nat. Mol. Cell. Biol. (2008), 9, 112-124). We have therefore employed a POPC bilayer here to facilitate comparison with experiment and because its homogeneity is advantageous for ensuring adequate configurational sampling during MD simulations.  We have revised and reordered the relevant text to put the discussion on membrane composition in one paragraph and we have added the following text (before l548 of the original manuscript) :

 “We have here simulated the proteins in a pure POPC bilayer. Phosphatidylcholine is the main lipid component of the mammalian ER membrane and POPC bilayers are often used as a simple ER mimic in in vitro studies. For example, we previously compared the heme tilt angle computed from simulations of three different CYPs in a POPC bilayer with that measured in experiments done on these CYPs in a Nanodisc containing a POPC bilayer and found very good agreement [24]. However, the ER membrane in fact contains a variety of glycerophospholipids, as well as cholesterol and ceramide. This more heterogeneous membrane composition may affect the protein positioning and dynamics as well as ligand entrance to the active site.”

Point:

Also, the 2.9 chapter can be combined with the discussion.

Response:

We agree that section 2.9 has a discussion component and therefore, we have changed the title of the ‘Results’ section (2) to ‘Results and Discussion’ and renamed ‘Discussion’ (3) to ‘Concluding Discussion’.

Point:

There are a few minor issues: in lines 477-480 a degree sign is not in superscript.

Response:

This has been corrected in the revised manuscript.

Point:

Also there is different spelling through the manuscript of enzymes: sometimes it is CYP 2C9 and sometimes CYP2C9.

Response:

We have revised the manuscript to ensure that the abbreviation of CYP 2C19 or CYP 2C9 is always used when referring to the respective cytochrome P450 proteins.  The gene names of CYP2C9 and CYP2C19 are only used when the gene is referred to, e.g. when referring to the isoforms of the human CYP2C family.

Point:

It would be good also in the main text to mention the quality of pdb models which were used to generate the starting models.

Response:

We have given the resolution of the PDB structures in the Methods section, “Preparation of structures of full-length models of CYP 2C9 and CYP 2C19”.